

# Multi-omic profiling to assess the effect of iron starvation in *Streptococcus pneumoniae* TIGR4

Irene Jiménez-Munguía[1], Mónica Calderón-Santiago[2], Antonio Rodríguez-Franco[1], Feliciano Priego-Capote[2] and Manuel J. Rodríguez-Ortega[1]

[1] Departamento de Bioquímica y Biología Molecular, Universidad de Córdoba; Campus de Excelencia Internacional CeiA3, Córdoba, Spain
[2] Departamento de Química Analítica, Universidad de Córdoba; Campus de Excelencia Internacional CeiA3, Córdoba, Spain

## ABSTRACT

We applied multi-omics approaches (transcriptomics, proteomics and metabolomics) to study the effect of iron starvation on the Gram-positive human pathogen *Streptococcus pneumoniae* to elucidate global changes in the bacterium in a condition similar to what can be found in the host during an infectious episode. We treated the reference strain TIGR4 with the iron chelator deferoxamine mesylate. DNA microarrays revealed changes in the expression of operons involved in multiple biological processes, with a prevalence of genes coding for ion binding proteins. We also studied the changes in protein abundance by 2-DE followed by MALDI-TOF/TOF analysis of total cell extracts and secretome fractions. The main proteomic changes were found in proteins related to the primary and amino sugar metabolism, especially in enzymes with divalent cations as cofactors. Finally, the metabolomic analysis of intracellular metabolites showed altered levels of amino sugars involved in the cell wall peptidoglycan metabolism. This work shows the utility of multi-perspective studies that can provide complementary results for the comprehension of how a given condition can influence global physiological changes in microorganisms.

## INTRODUCTION

Bacteria need a plethora of factors for optimal growth, with iron being an essential micronutrient. Within the human body, the free concentration of this element is approximately $10^{-18}$ M (*Yang et al., 2014*), which is too low to hold a growth capable of supporting bacterial infections. The low concentrations of free iron within the host are due to the scavenging of this element by different high-affinity proteins (e.g., transferrin, lactoferrin, haemoglobin, etc.) (*Froehlich, Bates & Scott, 2009*). Therefore, in order to survive in the host, pathogens need to develop special strategies to uptake the minimum amount that they require of such a nutrient (*Nanduri et al., 2008*), such as the direct extraction of this metal cation from host iron-containing proteins, or by capturing ferric-binding siderophores from host environments via ABC transporters (*Ge & Sun, 2014*).

Corresponding author
Manuel J. Rodríguez-Ortega,
mjrodriguez@uco.es

Moreover, the capacity of bacterial pathogens for iron acquisition itself represents an important virulence determinant (*Kanaujia et al., 2015*). In addition, pathogens can also modify their energetic metabolism to adapt to the environmental situation within the host.

*Streptococcus pneumoniae*, also known as the pneumococcus, is a Gram-positive microorganism that lives as a commensal in the human respiratory tract and that, under appropriate circumstances, becomes pathogenic, being able to cause high morbidity and mortality (*Blasi et al., 2012*; *Olaya-Abril et al., 2014b*). This bacterium is a major cause of mucosal diseases such as otitis media and sinusitis, and is a prevalent pathogen in different invasive diseases including pneumonia, bacteremia, meningitis, and sepsis (*O'Brien et al., 2009*). Pneumococcal pneumonia, which is the major clinical manifestation of pneumococcal infections, affects mainly young children and the elderly although all age groups may be affected. It has been estimated that almost one million children die every year because of pneumococcal diseases, with >90% of these deaths occurring in developing countries (*Johnson et al., 2010*). Pneumococcal infections also represent a high burden of disease in adults of developed countries. Actually, around 25,000 deaths are registered every year in the United States in adults >50 years of age, and in European countries the pneumococcus also causes significant mortality and morbidity (*Olaya-Abril et al., 2013*; *Weycker et al., 2010*). Therefore, understanding the basics of host-pathogen interactions is critical to effectively fight against infections.

In the context of systems biology, the use of massive analysis platforms is highly valuable to understand biological processes (*Fondi & Lio, 2015*; *Kohlstedt et al., 2014*). However, single-omic datasets, although powerful, offer a partial view of a biological system (*Grady et al., 2017*). Studying the responses of any biological system to a given condition using transcriptomics, proteomics and/or metabolomics can greatly help to elucidate the global adaptation to such a condition, i.e., stress, pathological status, nutrient availability/limitation, etc (*Dall'Agnol et al., 2014*; *Feng et al., 2011*; *Fu et al., 2013*; *Yang et al., 2012*). To this regard, we have approached from the in vitro bacterial culture to what it should actually occur in vivo, mimicking the iron restriction in order to describe the global changes that the pneumococcus undergoes and, therefore, to understand mechanisms of adaptation when it infects the host. To this end, we have studied the responses of the reference pneumococcal strain TIGR4 in an iron-deprived medium, at three different –omic levels: transcriptomics, proteomics and metabolomics. We have identified sets of genes, proteins and metabolites that are differentially expressed/synthesized under the studied nutrient restriction.

## MATERIALS AND METHODS

### Bacterial strains and culture conditions

*S. pneumoniae* TIGR4 was grown without agitation at 37 °C in air with 5% $CO_2$ in Todd Hewitt Broth (THB) until mid-exponential phase ($OD_{600} = 0.3$), and kept at −80 °C with 20% glycerol. Three different biological replicates were made for proteomics and metabolomics experiments, and four replicates for transcriptomics (20 mL, 100 mL, and 200 mL cultures to perform transcriptomics, metabolomics, and proteomics experiments,

respectively). Each set of replicates was standardised by inoculum, using starter cultures from glycerol-kept vials, in order to prepare the standard inoculum that was further added to the different biological replicates for each ''ome'' extraction. Iron-depleted cultures were prepared by adding deferoxamine (DFO) mesylate salt (Sigma) dissolved in water, at 100 μM, according to the dose used for this chelator in other works for the pneumococcus (*Trappetti et al., 2011*).

## Protein extracts

Total cellular proteins and secreted proteins were obtained as described (*Mitsuwan et al., 2017*). Briefly, to obtain cellular proteins, pellets were washed three times in sterile phosphate buffered saline (PBS) pH 7.4. Bacterial cell wall was digested at 37 °C with top-down agitation by adding 100 U mutanolysin (Sigma–Aldrich, St. Louis, MO, USA). Protoplasts were resuspended in 4 mL of rehydration buffer (7 M urea, 2 M thiourea, 4% CHAPS, 0.5% Triton X-100, 0.005% bromophenol blue, 0.5% Bio-lyte 3–10 ampholytes (Bio-Rad, Hercules, CA, USA)) and disrupted by sonication (6 cycles; 20-s pulses, 90% amplitude). Proteins were recovered by centrifugation (5,000× g, 7 min), dialyzed and concentrated using a centrifugal filter device (Amicon Ultra15, 10 kDa; Millipore, Burlington, MA, USA). To obtain secretome fractions, proteins were precipitated from the supernatants with 10% trichloroacetic acid, after removing cell debris through filter devices (0.22 μm, Millipore, Burlington, MA, USA). Protein pellets were washed twice with 1 mL of ice-cold absolute ethanol. Finally, proteins were air-dried and resuspended in 500 μL of rehydration buffer.

## Two-dimensional polyacrylamide gel electrophoresis and image analysis

Protein samples were cleaned using the 2-D clean up kit (Life Sciences, Marlborough, MA, USA) according to manufacturer's instructions. Proteins were resuspended in 200 μL of rehydration buffer and quantified by the Bradford method (*Bradford, 1976*). Five hundred μg of protein were subjected to isoelectric focusing (IEF) on 18 cm Immobiline DryStrips immobilized pH gradient (IPG) gel strips (4–7 pH linear gradient (Life Sciences, Marlborough, MA, USA)). The strips were loaded onto a Bio-Rad Protean IEF Cell system (Bio Rad, Hercules, CA, USA), and IEF was performed at 20 °C using the following conditions: 2 h of passive rehydration, 50 V for 10 h followed by a voltage-ramp (250 V for 15 min; 500 V for 30 min; 1,000 V for 1 h; 2,000 V for 1 h; 5,000 V for 1 h; 8,000 V for 2 h); finally, proteins were focused on 70,000 Vh. Before the second dimension, the IPG strips were first soaked for 15 min in equilibration solution (50 mM Tris- HCl buffer, pH 8.8, 6 M urea, 30% v/v glycerol, 2% SDS, and bromophenol blue traces) containing 2.5 mg/mL DTT, and subsequently soaked for 15 min in equilibration solution containing 45 mg/mL iodoacetamide. The second dimension was performed on 12% polyacrylamide gels, using the Protean plus Dodeca Cell system (Bio-Rad, Hercules, CA, USA). Gels were run at 90 V until the dye reached the bottom. Then, gels were stained with brilliant blue G-colloidal solution (Sigma-Aldrich, St. Louis, MO, USA) according to manufacturer's instructions. Gels were scanned with a GS-800 densitometer (Bio-Rad, Hercules, CA, USA). Digitized

images were analyzed with PD-Quest v8.1.0 (Bio-Rad, Hercules, CA, USA). Two analytical gels were made per sample (i.e., biological replicate and protein extraction). Consistent spots were considered as those whose presence remained constant at the three biological replicates. Spots showing a consistent change in the intensity value of at least twofold were included in the quantitative analysis.

## Protein identification by MALDI-TOF/TOF MS/MS

Spots excision, protein digestion, peptide desalting and mass spectrometry analysis were performed as already described by our group (*Mitsuwan et al., 2017*) with slight modifications: after mass spectra acquisition using a MALDI-TOF/TOF (4800 Proteomics Analyzer, Applied Biosystems, Foster City, CA, USA) mass spectrometer in the m/z range 800 to 4,000, Mascot 2.0 search engine (Matrix Science Ltd., London, UK) was used for protein identification running on GPS Explorer™ software v3.5 (Applied Biosystems, Foster City, CA, USA) over the National Center for Biotechnology Information (NCBI) protein database (updated monthly). Search setting allowed one missed cleavage with the selected trypsin enzyme, *Streptococcus pneumoniae* for taxonomy restrictions, cysteine carbamidomethylation as a fixed modification, methionine oxidation as a variable modification, a MS/MS fragment tolerance of 0.2 Da, and a precursor mass tolerance of 10 ppm. Identifications with a Mascot score >70 (*p*-value <0.05) were considered as significant.

## Inductively coupled plasma-mass spectrometry (ICP-MS) analysis

Iron concentration in THB medium was determined by ICP-MS. Five ml aliquots of samples were chemically digested with 2 mL concentrated nitric acid on a hot plate with a heating ramp of 20 °C until reaching 130 °C, maintaining this temperature for 2 hours. High purity deionized water was added to the digested samples to a final volume of 25 mL. Rh-solution was added as internal standard (final concentration of 10 μg/L). The isotope $^{56}$Fe and the internal standard were analyzed with a NexION 350X instrument (PerkinElmer, Waltham, MA, USA), equipped with a PFA concentric microFlow nebulizer and a cyclonic PFA spray chamber, and operated at 1,600 W in He collision mode. Results were expressed as μg of Fe per liter of culture.

## RNA isolation

Cells resuspended in 1 mL of Tri-Reagent (Sigma–Aldrich, St. Louis, MO, USA) were disrupted by vortexing (20 min) with 0.5 g of glass beads (Sigma–Aldrich, St. Louis, MO, USA). After recovering the supernatant by centrifugation (1 min, 12,000× g, 4 °C), 200 μL of chloroform were added. Samples were centrifuged again (12,000× g for 15 min; 4 °C) and 500 μL of ice-cold isopropanol were added. After 15 min incubation (4 °C), samples were centrifuged (30 min, 12,000× g, 4 °C) and washed with 500 μL of 70% ice-cold ethanol. RNA was air-dried and resuspended in 40 μL of distilled water previously treated with 1% of diethylpyrocarbonate (DEPC). Samples were treated with DNAse (Ambion, Austin, TX, USA) according to manufacturer's instructions.

## RNA amplification, labeling and hybridization to DNA microarrays

RNA quality was assessed using a TapeStation (Agilent Technologies, Santa Clara, CA, USA). The RNA integrity number (RIN) ranged between 7.0 and 9.2. Samples with RIN >7 were considered for analysis. RNA concentration and dye incorporation were measured using a UV–VIS spectrophotometer (Nanodrop 1000, Agilent Technologies, Santa Clara, CA, USA). Hybridization to custom 8 × 15 K Gene Expression Microarrays (ID 044371, Agilent Technologies, Santa Clara, CA, USA) containing the whole genome of *S. pneumoniae* TIGR4 was conducted following manufacturer's protocol using a two-color (Cy3 and Cy5) Microarray-Based Gene Expression Analysis (v. 6.5, Agilent Technologies, Santa Clara, CA, USA). Microarrays were then washed and scanned using a DNA Microarray Scanner (Model G2505C).

## Gene expression analysis

Microarray hybridization data were obtained with the Feature Extraction Software v. 10.7 (Agilent Technologies, Santa Clara, CA, USA), using the default variables. Data analysis was performed using the R Limma *Bioconductor* package (*Ritchie et al., 2015*), according to a direct two color design. A total of eight microarrays were done, corresponding to four biological replicates for each condition using swapped and random mixtures to cope with batch effects. Functional annotation of the differentially expressed genes was done using embedded BlastX included into the *Blast2GO* program (*Conesa et al., 2005*) using the public NCBI nr database. Raw feature intensities were corrected using the *Normexp* background correction algorithm. An initial within-array normalization was done using spatial and intensity-dependent *Loess* method, followed by a between-array *Aquantile* normalization. Normalized data are shown in Figs. S1 and S2. Differential expression was ordered according to their adjusted *p*-values, and the expression of each gene is reported as the $log_2$ ratio of the value obtained for each condition compared to control condition. A gene was considered differentially expressed if it displayed an adjusted *p*-value less than 0.05 by the Student *t*-test. Finally, over- and under- expressed genes were analyzed in terms of gene ontology by using a hypergeometric analysis (GOStats package). Prediction of operons was obtained from the DOOR database (*Dam et al., 2007*; *Mao et al., 2009*).

## Preparation of intracellular metabolite samples

Cell pellets were washed twice with PBS and resuspended in lysis buffer (PBS and 30% sucrose) containing 100 U mutanolysin. Samples were incubated at 37 °C overnight. Metabolic quenching was achieved by adding ice-cold 50% methanol. Cells were disrupted by sonication (6 cycles: 20 s, 90% amplitude). After centrifugation (5,000× g, 7 min), cell debris was separated through 0.22 µm membrane filters (Millipore, Burlington, MA, USA). Then, supernatants were collected and ultracentrifuged (100,000× g; 1.5 h, 4 °C). Finally, metabolite samples were kept at −80 °C before analysis.

## Analysis of intracellular metabolites

Intracellular metabolite samples were analyzed by LC–QTOF MS/MS using an Agilent 1200 Series LC system coupled to an Agilent 6540 UHD Accurate-Mass QTOF hybrid mass spectrometer equipped with dual electrospray (ESI) source as described (*Mitsuwan*

*et al., 2017*) without modifications. Briefly, chromatography was performed using a C18 reverse-phase analytical column (Mediterranean, 50 mm × 0.46 mm *i.d.,* 3 μm particle size; Teknokroma, Barcelona, Spain), thermostated at 25 °C. The mobile phases were 5% ACN (phase A) and 95% ACN (phase B) both with 0.1% formic acid as ionization agent. The LC pump was programmed with a flow rate of 0.8 mL/min with the following elution gradient: 3% phase B was kept as initial mobile phase constant from min 0 to 1; from 3 to 100% of phase B from min 1 to 13. A post-time of 5 min was set to equilibrate the initial conditions for the next analysis. The injection volume was 3 μL and the injector needle was washed for 10 times between injections with 80% methanol. The parameters of the electrospray ionization source, operating in negative and positive ionization mode, were as follows: the capillary and fragmentor voltage were set at ±3.5 kV and 175 V, respectively; $N_2$ in the nebulizer was flowed at 40 psi; the flow rate and temperature of the $N_2$ as drying gas were 8 L/min and 350 °C, respectively. MS and MS/MS data were collected in both polarities using the centroid mode at a rate of 2.6 spectra per second in the extended dynamic range mode (2 GHz). Accurate mass spectra in auto MS/MS mode were acquired in bot MS and MS/MS m/z ranges of 60–1,100 Da. To assure the desired mass accuracy of recorded ions, continuous internal calibration was performed during analyses by using the signals at m/z 121.0509 (protonated purine) and m/z 922.0098 (protonated hexakis (1H, 1H, 3H-tetrafluoropropoxy) phosphazine or HP-921) in positive ion mode; while in negative ion mode ions with m/z 119.0362 (proton abstracted purine) and m/z 966.0007 (formate adduct of HP-921) were used. The auto MS/MS mode was configured with two maximum precursors per cycle and an exclusion window of 0.25 min after two consecutive selections of the same precursor. The collision energy selected was 20 V.

## Metabolomics data processing and identification

MassHunter Workstation software (version 5.00 Qualitative Analysis, Agilent Technologies, Santa Clara, CA, USA) was used to process all data obtained by LC–QTOF in auto MS/MS mode. Treatment of raw data files was initiated by extraction of potential molecular features (MFs) with the suited algorithm included in the software. For this purpose, the extraction algorithm considered all ions exceeding 1,000 counts with a single charge state. Additionally, the isotopic distribution to consider a molecular feature as valid should be defined by two or more ions (with a peak spacing tolerance of 0.0025 *m/ z*, plus 10.0 ppm in mass accuracy). Adduct formation in the positive (+Na) and negative (+HCOO) modes, as well as neutral loss by dehydration were included to identify features corresponding to the same potential metabolite. Then, MFs, characterized by their retention time, intensity in the apex of chromatographic peak and accurate mass, were exported in compound exchange format files (.cef files) and imported into Mass Profiler Professional (MPP) software package (version 12.1, Agilent Technologies, Santa Clara, CA, USA) for alignment and further processing. MPP allowed statistical analysis by Volcano plot (a combination of analysis of variance and fold change analysis). The MS/MS METLIN Personal Compound and Database Library (PCDL) was used to identify significant compounds using both MS and MS/MS information to assure metabolite identification. To identify compounds with

no MS/MS information in the METLIN Database, MetaCyc database, and the MassBank database were used.

## Statistical analysis

All the quantitative analyses (transcriptomics, proteomics, and metabolomics) were performed from three or four independent biological replicates, and the results are expressed as the mean ± standard deviation. Paired data were analyzed by univariate analysis using the Student's $t$-test. Principal component analysis (PCA) was done with the web-based software NIA array analysis tool (http://lgsun.grc.nia.nih.gov/anova/index.html) (*Sharov, Dudekula & Ko, 2005*). $p$-Values lower than 0.05 were considered statistically significant.

## RESULTS

### Effect of iron starvation on pneumococcal growth *in vitro*

*S. pneumoniae* TIGR4 was grown in THB, in which the iron concentration available for the bacteria, measured by ICP-MS, was 734 µg/L. In order to perform the further transcriptomic, proteomic and metabolomic analyses, we firstly monitored the growth of this strain in the presence of 100 µM DFO. As observed for other microorganisms, iron depletion caused a slight lag phase in the growth of pneumococcus (control growth rate: $1.93\,h^{-1}$; DFO-treated culture growth rate: $1.39\,h^{-1}$), and instead lead to a decrease in the OD at the plateau phase (Fig. 1). We chose the mid-exponential phase for sampling RNA, proteins and metabolites in curve points such that almost no delays in the time points of collecting cultures took place. These corresponded to OD = 0.3.

### Transcriptomics

Limma analysis of the signals obtained after hybridizing the RNAs with the arrays revealed a number of differentially expressed genes when comparing the DFO treatment with its control (Supplemental Information 1): 338 genes changed significantly after DFO exposure, of which 118 genes increased their expression and 220 were down-regulated. Figure S3 shows the differentially expressed genes as a Volcano plot.

The $\log_2$ fold change values, resulting from averaging four independent biological replicates for each condition, ranged between 1.78 and −2.13. We noticed a consistent differential expression in which all of the genes pertaining to a same operon (according to the DOOR annotation database) changed in the same trend and even experienced almost identical $\log_2$ fold change values. We grouped the genes into operons according to the DOOR database (Supplemental Information 1). In total, 29 operons were clearly up-regulated, including those coding for the iron-compound ABC transporter system (genes SP_1869, SP_1870 and SP_1871), the closely neighbour gene SP_1872 coding for the iron-compound ABC transporter, the manganese ABC transporter system (genes SP_1648, SP_1649 and SP_1650), the ATP synthase complex (genes SP_1509 to SP_1513), and others including several transporter systems. Also, an operon containing genes for the riboflavin biosynthetic system (genes SP_0175 to SP_0178) was clearly up-regulated. Forty-four operons were clearly down-regulated, including the branched-chain amino

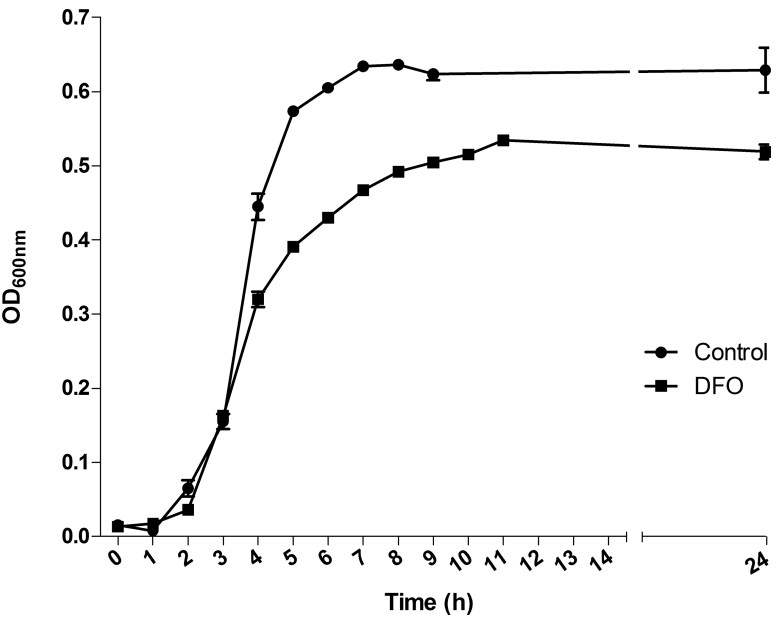

**Figure 1** **Effect of deferoxamine (DFO) on the growth of *Streptococcus pneumoniae* TIGR4.** Each determination represents the mean of three different biological replicates. Solid circles indicate the mean ± standard deviation (SD) of untreated cultures; solid squares indicate the mean ± SD of DFO-treated cultures.

acid transporter system (genes SP_0750 to SP_0753) and the large operon of 13 genes (SP_0419 to SP_0431) involved in fatty acid biosynthesis, as well as two operons involved in competence (genes SP_0042 and SP_0043, and genes SP_2235 and SP_2236). In all the cases, the log$_2$ FC values were almost identical for all the genes of the same operon. Table 1 shows the main operons that were differentially regulated.

The differentially expressed genes were annotated with Gene Ontology terms, using Blast2GO (Fig. 2). According to the molecular function level, the most enriched term among the over-expressed genes corresponded to "ion binding" (GO:0043167), representing around 25% of the genes. At the biological process level, the most enriched term was "cell metabolic process" (GO:0044237). For down-expressed genes, the most prevalent term at the biological process level was the same as for over-expressed genes, i.e., "cell metabolic process", and at the molecular function level, three different terms, including also that of "ion binding" (GO:0043167), represented around 13% each.

We validated the results obtained in the microarrays with RT-q-PCR on a subset of 16 differentially expressed genes of the TIGR4 strain (Table S1). For all the genes, there was the same trend (either over- or down-expression) and a high correlation ($r^2 = 0.93$) in the fold changes measured both at the microarray and the RT-q-PCR, thus confirming that the microarray data were reliable.

## Proteomics

We compared the proteomes of two different protein fractions, cell extracts and secreted proteins, by 2-DE under iron deprivation conditions (Fig. S4). We then analyzed both the

**Table 1  Operon distribution of differentially expressed genes under iron starvation.**

| ID Operon | Locus | Expressed genes | Change |
|---|---|---|---|
| 38501 | SP_0175, SP_0176, SP_0177, SP_0178 | 4/4 | Upregulated |
| 38508 | SP_0202, SP_0203 | 2/2 | Upregulated |
| 38509 | SP_0204, SP_0205, SP_0206, SP_0207 | 4/4 | Upregulated |
| 38541 | SP_0385, SP_0386 | 2/3 | Upregulated |
| 38567 | SP_0515, SP_0516 | 2/2 | Upregulated |
| 38588 | SP_0599, SP_0600, SP_0601 | 3/4 | Upregulated |
| 38589 | SP_0603, SP_0604 | 2/2 | Upregulated |
| 38594 | SP_0622, SP_0623, SP_0624 | 3/4 | Upregulated |
| 38595 | SP_0627, SP_0628 | 2/3 | Upregulated |
| 38600 | SP_0661, SP_0662 | 2/2 | Upregulated |
| 38642 | SP_0868, SP_0869 | 2/5 | Upregulated |
| 38672 | SP_0999, SP_1000 | 2/2 | Upregulated |
| 38738 | SP_1340, SP_1341, SP_1342, SP_1343, SP_1344 | 5/5 | Upregulated |
| 38776 | SP_1509, SP_1510, SP_1511, SP_1512, SP_1513 | 5/8 | Upregulated |
| 38789 | SP_1566, SP_1569 | 2/6 | Upregulated |
| 38795 | SP_1591, SP_1592 | 2/2 | Upregulated |
| 38805 | SP_1648, SP_1649, SP_1650 | 3/3 | Upregulated |
| 38808 | SP_1662, SP_1665 | 2/8 | Upregulated |
| 38809 | SP_1669, SP_1670 | 2/3 | Upregulated |
| 38833 | SP_1774, SP_1775 | 2/3 | Upregulated |
| 38839 | SP_1802, SP_1803, SP_1804 | 3/4 | Upregulated |
| 38854 | SP_1860, SP_1861, SP_1862, SP_1863 | 4/4 | Upregulated |
| 38857 | SP_1869, SP_1870, SP_1871 | 3/3 | Upregulated |
| 38862 | SP_1906, SP_1907 | 2/2 | Upregulated |
| 38884 | SP_2001, SP_2002 | 2/6 | Upregulated |
| 38896 | SP_2070, SP_2071 | 2/3 | Upregulated |
| 38917 | SP_2174, SP_2175, SP_2176 | 3/5 | Upregulated |
| 38922 | SP_2196, SP_2197 | 2/4 | Upregulated |
| 38930 | SP_2239, SP_2240 | 2/2 | Upregulated |
| 38468 | SP_0021, SP_0022 | 2/2 | Downregulated |
| 38469 | SP_0024, SP_0025, SP_0026 | 3/3 | Downregulated |
| 38472 | SP_0042, SP_0043 | 2/2 | Downregulated |
| 38476 | SP_0053, SP_0054, SP_0055 | 3/4 | Downregulated |
| 38486 | SP_0119, SP_0120 | 2/2 | Downregulated |
| 38495 | SP_0151, SP_0152 | 2/3 | Downregulated |
| 38498 | SP_0164, SP_0165 | 2/2 | Downregulated |
| 38510 | SP_0220, SP_0221, SP_0222 | 3/14 | Downregulated |
| 38529 | SP_0321, SP_0322, SP_0323, SP_0324, SP_0325, SP_0326 | 6/6 | Downregulated |
| 38533 | SP_0352, SP_0354 | 2/9 | Downregulated |
| 38546 | SP_0412, SP_0413 | 2/2 | Downregulated |

**Table 1** (*continued*)

| ID Operon | Locus | Expressed genes | Change |
|---|---|---|---|
| 38547 | SP_0416, SP_0417 | 2/2 | Downregulated |
| 38533 | SP_0352, SP_0354 | 2/9 | Downregulated |
| 38546 | SP_0412, SP_0413 | 2/2 | Downregulated |
| 38547 | SP_0416, SP_0417 | 2/2 | Downregulated |
| 38548 | SP_0419, SP_0420, SP_0421, SP_0422, SP_0423, SP_0424, SP_0425, SP_0426, SP_0427, SP_0428, SP_0429, SP_0430, SP_0431 | 13/13 | Downregulated |
| 38571 | SP_0526, SP_0527 | 2/3 | Downregulated |
| 38610 | SP_0701, SP_0702 | 2/2 | Downregulated |
| 38617 | SP_0737, SP_0738 | 2/2 | Downregulated |
| 38620 | SP_0750, SP_0751, SP_0752, SP_0753 | 4/4 | Downregulated |
| 38624 | SP_0768, SP_0770 | 2/4 | Downregulated |
| 38627 | SP_0785, SP_0786, SP_0787 | 3/3 | Downregulated |
| 38655 | SP_0918, SP_0919, SP_0920, SP_0921, SP_0922 | 5/5 | Downregulated |
| 38665 | SP_0963, SP_0964 | 2/2 | Downregulated |
| 38674 | SP_1013, SP_1014 | 2/2 | Downregulated |
| 38682 | SP_1069, SP_1070, SP_1071 | 3/6 | Downregulated |
| 38684 | SP_1079, SP_1080 | 2/2 | Downregulated |
| 38726 | SP_1276, SP_1277, SP_1278 | 3/3 | Downregulated |
| 38731 | SP_1294, SP_1295 | 2/3 | Downregulated |
| 38751 | SP_1402, SP_1404, SP_1405 | 3/4 | Downregulated |
| 38757 | SP_1428, SP_1429 | 2/2 | Downregulated |
| 38763 | SP_1460, SP_1461 | 2/2 | Downregulated |
| 38764 | SP_1462, SP_1463, SP_1464 | 3/3 | Downregulated |
| 38765 | SP_1465, SP_1466 | 2/2 | Downregulated |
| 38778 | SP_1522, SP_1523 | 2/7 | Downregulated |
| 38815 | SP_1686, SP_1687, SP_1688, SP_1689 | 4/4 | Downregulated |
| 38822 | SP_1724, SP_1725 | 2/2 | Downregulated |
| 38840 | SP_1809, SP_1810 | 2/2 | Downregulated |
| 38867 | SP_1920, SP_1922 | 2/3 | Downregulated |
| 38868 | SP_1923, SP_1924, SP_1925, SP_1926 | 4/4 | Downregulated |
| 38873 | SP_1948, SP_1949 | 2/2 | Downregulated |
| 38874 | SP_1951, SP_1952, SP_1953, SP_1954 | 4/5 | Downregulated |
| 38898 | SP_2085, SP_2086, SP_2087, SP_2088 | 4/4 | Downregulated |
| 38906 | SP_2115, SP_2117 | 2/3 | Downregulated |
| 38907 | SP_2118, SP_2119 | 2/3 | Downregulated |
| 38929 | SP_2235, SP_2236 | 2/3 | Downregulated |

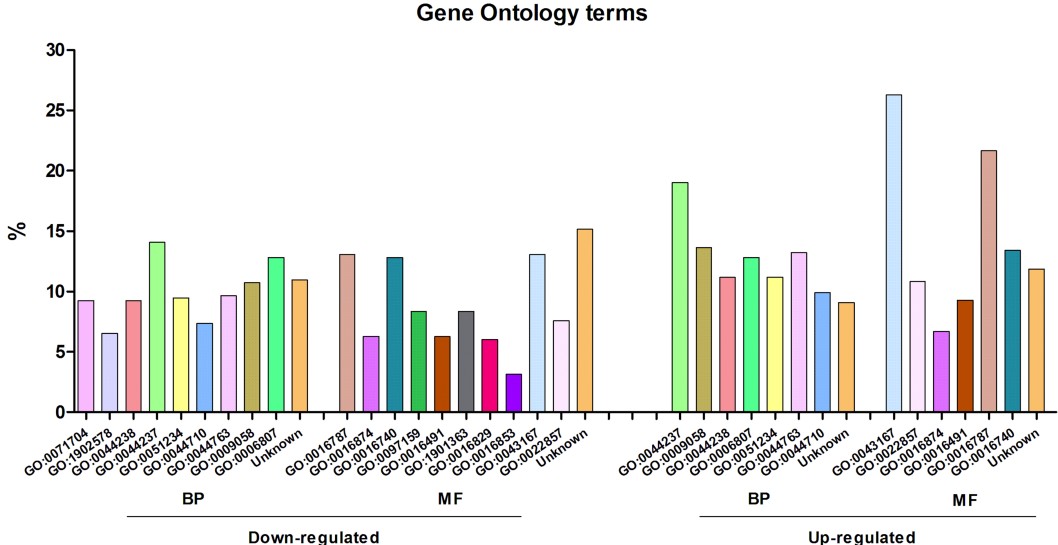

**Figure 2** **Gene Ontology analysis of the differentially expressed genes of *Streptococcus pneumoniae* TIGR4 after deferoxamine treatment.** Gene Ontology terms were retrieved from annotated genes or their orthologs in case of non-annotated genes using BLAST2GO. Terms are indicated in the GO nomenclature. BP, biological process; MF, molecular function.

qualitative (i.e., absolute appearance or disappearance of a protein in a condition compared to the other one) and quantitative changes (i.e., changes in protein abundance on spots present in both conditions) when comparing for each protein fraction the iron starvation condition with its non-iron deprived control (Tables 2 and 3).

For the two analyzed protein fractions, most of the changes corresponded to predicted cytoplasmic proteins, mainly affecting enzymes of the primary metabolism. Glyceraldehyde-3-phosphate dehydrogenase was less abundant under iron deprivation. This enzyme was strongly reduced in total extracts after DFO treatment (fold-change decrease > 4), as well as not detected in the secretome fraction of DFO-exposed culture (but detected in the control secretome). Another enzyme of the glycolysis pathway that also decreased its abundance was the phosphoglycerate kinase. Other two $Mg^{2+}$-dependent enzymes of the glycolysis pathway, enolase and pyruvate kinase, also decreased in secretomes of DFO-exposed cells. We also found that one enzyme of the fatty acid biosynthesis pathway was more abundant under iron deprivation: 3-oxoacyl-[acyl-carrier protein] reductase in total extracts (fold change around 7). Also, two major and very abundant extracellular proteins were detected to decrease in DFO-treated cultures: the choline-binding protein A (SP_2190) and the PcsB protein (SP_2216).

We also found a decrease in the abundance levels of an enzyme taking part in the recycling pathway of amino sugar compounds: the divalent cation-dependent *N*-acetylglucosamine-6-phosphate deacetylase, SP_2056 (NagA), which was found to be almost ninefold less abundant in DFO-exposed secretomes.

Other cation-containing or dependent enzymes decreased their abundances under iron deprivation conditions. Thus, lower levels of zinc-containing alcohol dehydrogenase

**Table 2** Protein identification by MALDI-TOF/TOF of qualitative changes in the protein abundances of *Streptococcus pneumoniae* TIGR4 after deferoxamine treatment.

| Spot | Abundance | Locus | Protein name | Score[a] | Location[b] |
|------|-----------|-------|--------------|----------|-------------|
| TE1 | Decreased | SP_1284 | LemA protein | 300 | Membrane |
| TE2 | Decreased | SP_1271 | 2-C-methyl-D-erythritol 4-phosphate cytidylyltransferase | 349 | Cytoplasmic |
| TE3 | Increased | SP_0845 | Basic membrane family protein | 942 | Extracellular |
| TE4 | Increased | SP_1362 | Glutamate dehydrogenase | 837 | Cytoplasmic |
| TS1 | Decreased | SP_1004 | Uncharacterized protein | 410 | Extracellular |
| TS2 | Decreased | SP_0341 | UPF0371 protein SP_0341 | 877 | Cytoplasmic |
| TS3 | Decreased | SP_0012 | Hypoxanthine-guanine phosphoribosyl-transferase | 749 | Cytoplasmic |
| TS4 | Decreased | SP_1229 | Formate–tetrahydrofolate ligase | 650 | Cytoplasmic |
| TS5 | Decreased | SP_0027 | Ribose-phosphate pyrophosphokinase 1 | 665 | Cytoplasmic |
| TS6 | Decreased | SP_0746 | ATP-dependent Clp protease proteolytic subunit | 81 | Cytoplasmic |
| TS7 | Decreased | SP_2190 | Choline binding protein A | 174 | Extracellular |
| TS8 | Decreased | SP_0715 | Lactate oxidase | 686 | Cytoplasmic |
| TS9 | Decreased | SP_2216 | Secreted 45 kd protein | 512 | Extracellular |
| TS10 | Decreased | SP_1456 | Peptide deformylase (PDF) | 251 | Cytoplasmic |
| TS11 | Decreased | SP_0459 | Formate acetyltransferase | 152 | Cytoplasmic |
| TS12 | Decreased | SP_1534 | Probable manganese-dependent inorganic pyrophosphatase | 533 | Cytoplasmic |
| TS13 | Decreased | SP_0516 | Protein GrpE (HSP-70 cofactor) | 178 | Cytoplasmic |
| TS14 | Decreased | SP_2012 | Glyceraldehyde-3-phosphate dehydrogenase | 451 | Cytoplasmic |
| TS15 | Decreased | SP_1445 | GMP synthase [glutamine-hydrolyzing] | 676 | Cytoplasmic |
| TS16 | Decreased | SP_0236 | DNA-directed RNA polymerase subunit alpha (RNAP subunit alpha) | 152 | Cytoplasmic |
| TS17 | Decreased | SP_1922 | Probable transcriptional regulatory protein SP_1922 | 418 | Cytoplasmic |
| TS18 | Decreased | SP_0435 | Elongation factor P (EF-P) | 290 | Cytoplasmic |
| TS19 | Decreased | SP_2084 | Phosphate-binding protein PstS 2 (PBP 2) | 862 | Extracellular |
| TS20 | Decreased | SP_1999 | Catabolite control protein A | 937 | Cytoplasmic |
| TS21 | Decreased | SP_1508 | ATP synthase subunit beta | 261 | Cytoplasmic |
| TS22 | Decreased | SP_0745 | Uracil phosphoribosyltransferase | 702 | Cytoplasmic |
| TS23 | Increased | SP_0862 | Ribosomal protein S3 | 424 | Cytoplasmic |

**Notes.**
[a] Mascot scores >70 were considered significant at $p < 0.05$.
[b] Protein localization in subcellular compartments was assigned using LocateP.

and manganese-dependent inorganic pyrophosphatase were detected, as well as of hypoxanthine-guanine phosphoribosyltransferase, which requires $Mg^{2+}$ as a cofactor.

## Metabolomics

Finally, we studied the changes in the metabolic profile of intracellular metabolite fractions from both pneumococcal strains, by using LC–MS/MS, which allowed detection of 719 and
**Table 3 Protein identification by MALDI-TOF/TOF of quantitative changes (>2 fold change (FC)) in the protein abundances of *Streptococcus pneumoniae* TIGR4 after deferoxamine treatment.**

| Spot | Abundance | FC[a] | *p*-value | Locus | Protein name | Score[b] | Location[c] |
|------|-----------|-----|-----------|-------|--------------|-------|-------------|
| TE5 | Increased | 6.1 | 0.03 | SP_1220 | L-lactate dehydrogenase | 607 | Cytoplasmic |
| TE6 | Increased | 6.9 | 0.01 | SP_0421 | 3-oxoacyl-[acyl-carrier protein] reductase | 528 | Cytoplasmic |
| TE7 | Decreased | 4.6 | 0.04 | SP_2055 | Alcohol dehydrogenase, zinc-containing | 535 | Cytoplasmic |
| TE8 | Decreased | 4.3 | 0.03 | SP_2012 | Glyceraldehyde-3-phosphate dehydrogenase | 255 | Cytoplasmic |
| TS24 | Increased | 4.4 | 0.02 | nanA | Sialidase A | 185 | Cell wall |
| TS25 | Decreased | 4.2 | 0.02 | SP_0499 | phosphoglycerate kinase | 991 | Cytoplasmic |
| TS26 | Decreased | 4.5 | 0.04 | SP_1489 | Elongation factor Tu | 1,120 | Cytoplasmic |
| TS27 | Decreased | 6.6 | 0.03 | SP_0897 | Pyruvate kinase | 708 | Cytoplasmic |
| TS28 | Decreased | 8.8 | 0.01 | SP_2056 | N-acetylglucosamine-6-phosphate deacetylase | 587 | Cytoplasmic |
| TS29 | Decreased | 4.2 | 0.03 | SP_1128 | Enolase | 349 | Cytoplasmic |
| TS30 | Decreased | 4.1 | 0.04 | SP_1541 | 30S ribosomal protein S6 | 114 | Cytoplasmic |
| TS31 | Decreased | 4.4 | 0.04 | SP_1910 | Uncharacterized protein | 523 | Cytoplasmic |

Notes.
[a]For increased proteins, the FC was calculated as the treatment/control ratio; for decreased proteins, the FC was calculated as the control/treatment ratio.
[b]Mascot scores >70 were considered significant at $p < 0.05$.
[c]Protein localization in subcellular compartments was assigned using LocateP.

826 different chromatographic peaks, in negative and positive ionization mode, respectively. Statistical analysis by Volcano plot revealed that 64 entities presented a *p*-value below 0.05 and a fold change, in terms of relative concentrations, higher than 2 for discrimination between treated and non-treated TIGR4 samples (data not shown).

A multivariate statistical analysis was carried out with significant entities to evaluate whether the iron deprivation treatment had an effect on the metabolite profile. Figure S5 shows the principal component analysis (PCA) of identified metabolites when comparing iron-depleted and non-depleted cultures, revealing that the first principal component (*X*-axis) clearly grouped the three control biological replicates, which were clearly separated from two out of the three DFO-treated samples. However, there was dispersion in the three biological replicates of the DFO treatment, as the first principal component did not group them.

The tentative identification of significantly changing entities led to a panel of 17 compounds (Table 4). Although the number of changing metabolites positively identified was low, we clearly found an increase in the concentration of intermediate amino sugar metabolites involved in the cell wall peptidoglycan metabolism: there was an increase in uridine-5′-diphosphoglucuronic acid (2.2-fold), UDP-*N*-acetylmuraminate (>20-fold), *N*-acetylglucosamine (3.8-fold), and UDP-*N*-acetylglucosamine (2.4-fold).

## DISCUSSION

There is little knowledge about the mechanisms of iron intake by the pneumococcus (*Hoyer et al., in press*). In this work, we have approached a multi-omics strategy to understand the changes at the molecular level occurring in the pneumococcus during iron deprivation, similarly to what theoretically happens during an *in vivo* infection. To our knowledge, this is a unique study carried out so far for iron deprivation in this human pathogen using

**Table 4  Metabolites altered after deferoxamine treatment of *Streptococcus pneumoniae* TIGR4.**

| Metabolite | Retention time (min) | Abundance | Fold change[a] | *p*-value |
|---|---|---|---|---|
| C-di-AMP | 5.86 | Increased | 3.9 | 0.02 |
| Uridine-5′-diphosphoglucuronic acid | 6.22 | Increased | 2.2 | 0.03 |
| UDP-N-acetylmuraminate | 1.50 | Increased | 20.2 | 0.01 |
| UDP-N-acetylglucosamine | 2.02 | Increased | 2.4 | 0.04 |
| cAMP | 5.94 | Increased | 3.4 | 0.03 |
| Tri-N-acetylchitotriose | 6.04 | Increased | 2.3 | 0.03 |
| Purine | 2.37 | Increased | 10.1 | 0.01 |
| dTMP | 5.98 | Decreased | 12.6 | 0.01 |
| Adenosine 5-monophosphate | 5.80 | Decreased | 17.9 | 0.01 |
| GDP-glucose | 6.44 | Decreased | 2.5 | 0.04 |
| GDP | 3.04 | Decreased | 2.1 | 0.03 |
| deoxyAMP | 5.06 | Decreased | 10.5 | 0.01 |
| Guanine | 5.96 | Decreased | 8.5 | 0.01 |
| deoxyGMP | 5.88 | Decreased | 12.0 | 0.01 |
| GluMet | 6.16 | Decreased | 2.4 | 0.03 |
| Metionine | 6.16 | Decreased | 2.4 | 0.02 |
| N-acetylglucosamine | 6.29 | Decreased | 3.8 | 0.02 |

**Notes.**

[a] For increased metabolites, the FC was calculated as the treatment/control ratio; for decreased metabolites, the FC was calculated as the control/treatment ratio.

three different "omics". Very recently, a combined translatomics/proteomics approach has been applied to identify novel iron-transporting proteins in the pneumococcus (*Yang et al., 2016*). Previously, it had been approached using only proteomics (*Nanduri et al., 2008*; *Yang et al., 2015*).

We selected as iron chelator the DFO, which has been described to have high iron chelating specificity, although it might also sequester other divalent cations (*Eichenbaum, Green & Scott, 1996*). We chose the culture points at which almost no alterations in growth were observed, as described for other studies in bacteria in which this iron chelator was used at the same or very similar concentration (*Basler et al., 2006*; *Smith et al., 2001*; *Trappetti et al., 2011*). These conditions were applied to obtain transcriptome, proteome and metabolome preparations of the reference pneumococcal strain TIGR4.

Our transcriptomic analysis showed very consistent and reproducible results between biological replicates. The range of $\log_2$ fold changes was apparently low, ranging between 1.78 and −2.13, but the numbers obtained were in general quite similar to those observed for transcriptomic analyses of iron starvation in other microorganisms (*Allen et al., 2010*; *Basler et al., 2006*; *Brickman et al., 2011*; *Klitgaard et al., 2010*; *Madsen et al., 2006*). However, very interestingly the changes observed in our work were in most cases for genes grouped in operons, and genes belonging to the same operon underwent, as expected, almost identical fold change values. This is another argument that confirms the validity of our microarray results, as already described in similar works (*Allen et al., 2010*; *Klitgaard et al., 2010*; *Vasileva et al., 2012*).

In TIGR4, we found an up-regulation of 29 operons, and a down-regulation of 44 operons. Among those being over-expressed, we found two operons coding for iron-compound ABC transport systems, both of them localized together in the genome: the operon 38857 (genes SP_1869 to SP_1871) and the operon 1446903 (containing only one gene, SP_1872). The TIGR4 genome has other operon (no. 38677) containing four genes (SP_1032 to SP_1035) participating in a third iron-compound ABC system, but this operon was not found differentially expressed in our study. The up-regulation of the manganese ABC transporter system (operon 38805) could indicate that the chelator used is not completely specific for iron, as already known, but it cannot be ruled out that this operon might have a function related to iron uptake. Actually, recently *Yang et al. (2016)* have reported the over-expression in the *S. pneumoniae* D39 strain of genes coding for sugar and other substrate-ABC transporters and validated one of them at the protein level, thus indicating that iron deprivation may affect other transporter systems which are not annotated in databases as "iron transporters". We also found the up-regulation of the operon 38501, responsible for riboflavin biosynthesis. Very recently, it has been described in *Vibrio cholerae* the cross-modulatory effect between riboflavin and iron (*Sepulveda-Cisternas et al., 2018*). Among the down-expressed operons, we found some transporter systems, as the branched-chain amino acid ABC transporter system (operon 38620, genes SP_0750 to SP_0753), a fluoride ion transporter system (genes SP_1294 and SP_1295) or a phosphate ABC transporter system (genes SP_2085 to SP_2088). We ignore the meaning of these changes, but this work opens new possibilities to explore the role of these genes in the iron uptake.

In previous works, we have analyzed the surface proteome ("surfome") of the pneumococcus specifically targeting the discovery of vaccine or diagnostic candidates (*Jimenez-Munguia et al., 2015*; *Olaya-Abril et al., 2012*; *Olaya-Abril et al., 2013*; *Olaya-Abril et al., 2015*). In this study, we searched for proteome changes using 2-D gel-based analysis on total cell extract and secretome fractions. As expected, most changes were in cytoplasmic proteins, as these are the most abundant ones in the bacterial cells. This class of proteins was also found in the secretome fractions, as extensively reported for numerous works in a wide variety of microorganisms (for an extensive review about the presence and role of cytoplasmic proteins in extracellular and/or surface protein preparations, see *Olaya-Abril et al., 2014a*).

Many of the changes observed in our work are coinciding with other results already described in the pneumococcus and other microorganisms. In the present study, the enzyme GAPDH was less abundant in iron-depleted protein fractions, as reported for pneumococcus (*Nanduri et al., 2008*) and *Staphylococcus aureus* (*Friedman et al., 2006*). This protein has been described as a moonlighting protein with different functions, including an important role in iron metabolism (*Boradia, Raje & Raje, 2014*). The zinc-containing alcohol dehydrogenase has been also described to be less abundant in iron-depleted *S. aureus* protein fractions (*Friedman et al., 2006*). We found a decrease in some $Mg^{2+}$-dependent glycolytic enzymes (phosphoglycerate kinase, enolase, pyruvate kinase), which might be due to the partially non-specific sequestration of this cation by DFO. Very

similar results have been obtained in pneumococcal biofilms, where iron is less available than in planktonic cultures (*Allan et al., 2014*; *Trappetti et al., 2011*).

In a very recent paper, Hoyer and colleagues have studied the changes in the proteome of *Streptococcus pneumoniae* D39 under iron deprivation using 2,2′-bipyridine as chelator, in two different culture media (*Hoyer et al., in press*). They have performed the proteomic analysis by LC-MS/MS, which is much more sensitive than our 2-DE/MALDI-TOF approach for detecting both high numbers of proteins and changes in their abundances. However, in spite of using different strains, growth media and proteomic approaches, by comparing our results with those obtained in the THY medium in this cited work, there is a strong coincidence in the protein changes: 16 out of our 38 changing different proteins (namely SP_1284, SP_2190, SP_1456, SP_0459, 1534, SP_2012, SP_0236, SP_1508, SP_0421, SP_2055, SP_0499, SP_1489, SP_0897, SP_2056, and SP_1128) changed accordingly in both works. Many of the proteins are cation-dependent, as already described above. Interestingly, one of the common changes is the NagA.

Perhaps metabolomics data are generally the most difficult ones to integrate with the other "omes", as it is still a step backwards compared to transcriptomics and proteomics. There are very few studies on metabolomics in bacteria. We have recently published a work of proteomics and metabolomics integration in the pneumococcus to study the effect of an antimicrobial compound (*Mitsuwan et al., 2017*). Very recently, Leonard and colleagues have described the metabolome inventory of a non-encapsulated *S. pneumoniae* TIGR4 in a chemically-defined medium, and using three techniques: $^1$H-NMR, HPLC-MS and GC-MS (*Leonard et al., in press*). Although these results are not comparable to ours (different growth medium, different methods and purpose), the cited work reveals the identification of some tens of metabolites, with a predominance of precursors of peptidoglycan synthesis (UDP-MurNAc, UDP-GlcNAc) as in our study. In this present work, we unambiguously detected only a few compounds changing after DFO treatment. Partially, this could be due to the variability in the three DFO-treated metabolome samples, as observed in Fig. S5. In addition, we could not calculate the adenylate energy charge, a well accepted quality control in metabolomic preparations, as we did not unambiguously identify ATP in our samples. These two issues indicate that our metabolomic data should be interpreted with caution. Nevertheless, some of the changes detected were in metabolites related to peptidoglycan metabolism. Very interestingly, the enzyme NagA, taking part in the amino sugar metabolism, was clearly depleted. Actually, according to the UniProtKB database, NagA has divalent cations as cofactors, including $Fe^{2+}$, as it possesses a conserved domain belonging to the metallo-dependent hydrolases superfamily. NagA deacetylates GlcNAc-6P to GlcN-6P. The increase observed in our work in GlcNAc, a precursor of GlcNAc-6P, might be due to the decrease in enzymes like NagA that avoid their by-products conversions. This protein has been recently detected in the pneumococcal secreted fraction as in our work, and has been proposed as a pneumococcal diagnostic marker because of its high immunogenicity (*Choi et al., 2013*).

In our opinion this work shows a possible unknown effect of iron deprivation on the global physiology of the bacterium, as it seems to be a relationship among iron starvation, the depletion of this enzyme and the alteration of the above discussed metabolites. Further

research will be needed to go in depth in this aspect. This may help to identify pathways or biomolecules that can be used as targets for therapies against pneumococcal infection.

## CONCLUSIONS

A global multi-omic analysis has been carried out to study the effect of iron limitation in the pneumococcus, similarly to what occurs during in vivo infection within the host. A significant number of genes changed in their expression, many of them involved in ion binding functions. Proteomics revealed changes in enzymes participating in the primary and peptidoglycan metabolism, many of them being cation-dependent. The metabolomic analysis revealed some changes in the levels of intermediates involved in the peptidoglycan biosynthesis. The different ''omics'' show sets of changing biomolecules that can complement themselves to provide a global insight into the adaptation of pneumococcus to iron starvation.

## ACKNOWLEDGEMENTS

MALDI-TOF/TOF and ICP-MS analyses were performed at the Proteomics and the Mass Spectrometry Facilities, respectively (SCAI, University of Córdoba). We are indebted to members of the AGR-164 group, University of Córdoba, for lab support.

### Funding

This research was funded by Project Grants FIS-P12/01259 (Spanish Ministry of Economy and Competitiveness), P09-CTS-4616 from Consejería de Innovación, Ciencia y Empresa (Junta de Andalucía), to Manuel José Rodríguez-Ortega, and by FEDER funds from the EU. The funders had no role in study design, data collection and analysis, decision to publish, or preparation of the manuscript.

### Grant Disclosures

The following grant information was disclosed by the authors:
Spanish Ministry of Economy and Competitiveness: FIS-P12/01259.
Consejería de Innovación, Ciencia y Empresa: P09-CTS-4616.
FEDER.

### Competing Interests

The authors declare there are no competing interests.

### Author Contributions

- Irene Jiménez-Munguía conceived and designed the experiments, performed the experiments, analyzed the data, prepared figures and/or tables, authored or reviewed drafts of the paper, approved the final draft.
- Mónica Calderón-Santiago performed the experiments, authored or reviewed drafts of the paper, approved the final draft.

- Antonio Rodríguez-Franco analyzed the data, contributed reagents/materials/analysis tools, prepared figures and/or tables, authored or reviewed drafts of the paper, approved the final draft.
- Feliciano Priego-Capote analyzed the data, contributed reagents/materials/analysis tools, authored or reviewed drafts of the paper, approved the final draft.
- Manuel J. Rodríguez-Ortega conceived and designed the experiments, analyzed the data, contributed reagents/materials/analysis tools, prepared figures and/or tables, authored or reviewed drafts of the paper, approved the final draft.

## Data Availability

The complete design of the microarray was deposited at the Gene Expression Omnibus NCBI database, with accession number GSE109693 (https://www.ncbi.nlm.nih.gov/geo/query/acc.cgi?acc=GSE109693).

## Supplemental Information

Supplemental information for this article can be found online at http://dx.doi.org/10.7717/peerj.4966#supplemental-information.

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
