# Peer review of "Multi-omic profiling to assess the effect of iron starvation in Streptococcus pneumoniae TIGR4"

_PeerJ, doi:10.7717/peerj.4966_

## Round 0.1 · original submission · Major Revisions

· Academic Editor

Major Revisions

The reviewers have analyzed the manuscript and although they have found merit there are several concerns. Both feel that recent data from the literature should be discussed, the experimental procedures needs clarification and a better integration of the data is necessary. If you consider that you can answer these concerns, we would be happy to receive a revised manuscript.

Reviewer 1 ·

Basic reporting

In this study the authors used multi omics strategies to investigate the effect of iron starvation on pneumococci.

Experimental design

• the authors mentioned that pneumococci were cultured in THB. Was this soley THB or was THB supplemented with yeast extract
• the authors used desferoxamine meslate salt for iron starvation. How was the iron concentration in THB with and without DFO determined. Which concentration is available for pneumococci in THB

Validity of the findings

• DFO effects on growth were tested using THB. The statement is that there was iron starvation did not cause a lag phase. However, when looking at the growth curves there seems to be a difference in growth and it is not clear whether this is a significant effect. Here, the values of the growth rates will be helpful
• Transcriptomics: the authors used biological replicates and performed a MA plot analysis (after normalization of the data). The plot suggests that the variance of the individual transcriptomic experiments is high and that a high number of genes in the plot biological replicates show high standard deviations and are out of the range for replicates. Please comment on the reproducibility of the microarray analysis. This is in particular important because of the data shown in Figure 2 and Table 1 as well as Table S1. In the latter table only a few genes are depicted (see also text line 263...) which do not reflecting the number of regulated genes described before.
• The proteome analysis was done by 2-D-GE. There is a recent paper (doi: 10.1016/j.ijmm.2018.02.001. [Epub ahead of print] PMID: 29496408) that has demonstrated the effects of iron limitation on the pneumococcal proteome using gel free LC MS/MS analysis. Here different conditions and iron concentrations were used. The effects of iron starvation on the cell phasiology and cell division have been demonstrated. I recommend a comparison of the datasets and to include this study in the paper. This will significantly strengthen the submitted study.
• How many replicates have been done for the 2-D-GE?
• The MS analysis has been done with fixed modification of methionine. Why not with variable modifications?
• Regarding the metabolome analysis the energy charge of the samples has not been mentioned in the methods. This is crucial for the reproducibility of the intracellular metabolome data under the selected conditions.
• Finally, a data integration approach would be advantageous, if possible

Additional comments

This is a study supporting earlier studies using a combined apporahc of multi-omics to demonstrate the physiological changes of pneumococci under iron starvation conditions

Reviewer 2 ·

Basic reporting

The structure of the manuscript follows that required in PeerJ. Overall, the manuscript is written to a respectable standard although further detailed editing is required throughout the manuscript to ensure consistency in quality.
Introduction presents the context of the subject of this work and the necessity of inclusion of physiologically relevant conditions into experimental design. The focus is towards cation limitation – namely iron – and its effects upon a representative strain of the major human pathogen Streptococcus pneumoniae.
Some specific background points lack suitable citations or are inaccurate. For instance, it is stated in lines 42-44 that “bacteria can express certain proteins called siderophores, specialized in uptaking divalent cations from the environment”. Firstly, siderophores are more typically secondary metabolite rather than proteins and – in respect of iron – are typically involved in acquisition of trivalent ions i.e. ferric iron (Fe3+). Does S. pneumoniae synthesise a siderophore or does it “scavenge” via other siderophores as well as heme? Inclusion of a brief consideration of iron acquisition systems specific to S. pneumoniae would be appropriate, e.g. Ge & Sun 2014.
There are two recent publications which are relevant to cite. These separately address metabolomics [Leonard et a., 2018] and proteomics [Hoyer et al., 2018] of S. pneumoniae under iron limitation. These will also be useful material for inclusion in Discussion.
In common with many presentations of omics data, summary Figures are provided in the text along with extended tables. Although not visually engaging, it is necessary and appropriate to provide summary data and to present results in tabular format. Among the figures, Supplementary Figure S1 in particular requires clarification: panel A indicates 8 samples termed A to H, each with a different colour line however only one line appears in the density x intensity plot (panel A). This would seem correct assuming plots in panel B are correct. Panel B presents an array x intensity plot which show appears to show identical range of values across all 8 arrays. Is this correct? A re-evaluation and inclusion of extended legend would be helpful.
There is an extended table with processed data for microarray experiment as Supplementary Dataset and a URL for GEO database entry is indicated.

Experimental design

The manuscript describes a combination of omics approaches to a representative of a species of significant human pathogens. The fundamental question and scientific approach fall within the scope of the journal.
The central purpose is to define alterations in expression (transcript, protein, metabolite) occurring in response to iron limitation. This is a physiologically-relevant stress and mimics a key aspect in nutrient limitation occurring in tissues. Iron limitation is a widely applied condition for exploration of bacterial characteristics resembling in vivo and has been used with many bacterial, including Streptococci. Two recent publications apply proteomics and metabolomics to S. pneumoniae and require consideration.
Experimentation employs a single bacterial isolate representing a high virulent clade of S. pneumoniae. This strain is widely used by many researchers for a range of purposes and does not pose ethical issues.
The microbiological methods, though routine, are not entirely clear. Standardisation is highly important given the sensitivity of omics methods to experimental variation. Some further detail of culture procedures would be helpful. For example, were replicate cultures standardised by inoculum; was a “starter” culture prepared then a standard inoculum dosed into experimental cultures? Was culture carried out with shaking or static; was this in air, with added carbon dioxide or other atmosphere? Was the desferoxamine concentration based on titration carried out by authors or based on previous publications? Optical density at time of sampling is presented in Results.

Validity of the findings

The combination of transcriptomics, proteomics and metabolomics are presented and analysed individually rather than integratively. The authors have previously carried out integration of proteomics and metabolomics data in a separate studiy with S. pneumoniae (Mitsuwan et al.,2017) and this could be a basis to extent interpretation in this manuscript.
Conclusions are broad and based around the combined omics approach and the specific issue of iron limitation on S. pneumoniae.
As noted in “Experimental Design” section, standardisation is particularly important in omics experimentation. Replicates for Controls and Test samples are included and these are indicated as “biological replicates”. Can the authors verify that ,in the context of this manuscript, this multiple cultures used as experimental replicates within each the omics studies.
Metabolomics data presentation (Supplementary Figure S5) draws attention to experimental reproducibility. The PCA plot shows that Control replicates were closely spaced, all falling in to single quadrant in the plot of PC1 and PC3. Why was PC3 selected rather than PC2 which would, presumably, account for a greater proportion of sample variability? Regarding the three iron-limited samples, each one falls into distinct quadrants in the PCA plot, suggesting a large extent of within-experiment variation. This variation presumably contributed to low numbers of consistently assigned metabolites. The authors should consider further the value of these analyses.
Discussion and Conclusions should take into account recent publications on proteomics and metabolomics of S. pneumoniae under distinct iron limitation conditions [Hoyer et al., 2018; Leonard et al., 2018] to broaden relevance.
Much of the Discussion is speculative in regard of function relevance of observed changes in transcript/protein/metabolite levels. The final sentence in Introduction states the “[this study] may help to identify pathways or biomolecules that can be used as targets for therapies against pneumococcal infection” and this should be revisited towards end of Discussion.

Additional comments

No additional comments

---

## Round 0.2 · Minor Revisions

· Academic Editor

Minor Revisions

The reviewers have analyzed the revised manuscript and considered that it has been considerably improved. However, reviewer 2 still has some concerns on the variability of the metabolomics. If you can address these issues, we would be glad to receive a revised manuscript.

Reviewer 1 ·

Basic reporting

na

Experimental design

na

Validity of the findings

na

Additional comments

The reviewer has no further comments. The authors have adequately responded to the comments of the reviewers and improved significantly the manuscript.

Reviewer 2 ·

Basic reporting

No additional comments.

Experimental design

No additional comment.

Validity of the findings

The variability in metabolomics results remains a concern although not subject to over-interpretation.

---

## Round 0.3 · accepted · Accept

· Academic Editor

Accept

I am happy to inform you that your manuscript has now been accepted for publication in PeerJ.

#